# Epstein–Barr Virus B Cell Growth Transformation: The Nuclear Events

**DOI:** 10.3390/v15040832

**Published:** 2023-03-24

**Authors:** Bo Zhao

**Affiliations:** Department of Medicine, Division of Infectious Diseases, Brigham and Women’s Hospital and Harvard Medical School, 181 Longwood Avenue, Boston, MA 02115, USA; bzhao@bwh.harvard.edu

**Keywords:** EBV, EBNA, LMP1

## Abstract

Epstein–Barr virus (EBV) is the first human DNA tumor virus identified from African Burkitt’s lymphoma cells. EBV causes ~200,000 various cancers world-wide each year. EBV-associated cancers express latent EBV proteins, EBV nuclear antigens (EBNAs), and latent membrane proteins (LMPs). EBNA1 tethers EBV episomes to the chromosome during mitosis to ensure episomes are divided evenly between daughter cells. EBNA2 is the major EBV latency transcription activator. It activates the expression of other EBNAs and LMPs. It also activates MYC through enhancers 400–500 kb upstream to provide proliferation signals. EBNALP co-activates with EBNA2. EBNA3A/C represses CDKN2A to prevent senescence. LMP1 activates NF-κB to prevent apoptosis. The coordinated activity of EBV proteins in the nucleus allows efficient transformation of primary resting B lymphocytes into immortalized lymphoblastoid cell lines in vitro.

## 1. Introduction

Epstein–Barr virus (EBV) was discovered from African Burkitt’s lymphoma (BL) cells over fifty years ago [1]. EBV causes many different cancers including B cell lymphomas and Hodgkin’s lymphoma in people with T-cell immune suppression due to HIV infection, transplantation, or advanced age; nasopharyngeal carcinoma; and ~10% of gastric cancers [2]. During primary EBV infection, viruses replicate in oro-pharyngeal epithelial cells and infect resting B-lymphocytes (RBLs). Latency III EBV nuclear antigens, EBNA2, leader protein (EBNALP), EBNA3A, EBNA3C, and Latent Membrane Protein 1 are the principal causes of infected B cell proliferation. While immune responses eliminate most B cells expressing latency III proteins, escaped cells enter lymphatic tissues, where EBV protein expression is down-regulated and only EBNA1 is expressed. EBV infected cells that escape immune destruction are sites of long-term latency persistence, reactivation, and can evolve to EBV+ B cell lymphomas and Hodgkin’s lymphoma [3].

EBV immortalizes RBLs to lymphoblastoid cell lines (LCLs) in vitro. LCLs express type III EBV latency genes. Post-transplant lymphoproliferative disease (PTLD) and AIDS CNS lymphomas express the same latency III EBV genes as LCLs. Therefore, LCLs are a useful and appropriate model for genetic and biochemical investigation of EBV’s roles in converting RBLs to proliferating lymphoblasts in vivo.

## 2. EBV Transformation from RBL to LCL Timeline

EBV infection of RBLs in vitro results in the establishment of LCLs in 3–4 weeks. Once established, LCLs can be kept in culture and grow continuously. Two days after EBV infection of RBLs, the cell size starts to increase and continues to increase. By day 4, cell size reaches the peak and starts to decrease slowly. Eight days after infection, the cell size is approximately two times the size of RBLs [4,5]. EBNA mRNAs are expressed at LCL level by day 2 and maintain a similar level afterwards. LMP1 mRNA level starts to increase two days after infection and gradually increases to LCL level at day 28 [6,7,8]. EBNA2 protein expression level reaches the peak level two days post infection and then slowly decreases to the LCL expression level [9]. EBNA3A and 3C protein expression start at day two, but only reach high level at day 4 and maintain the expression at that level [9]. LMP1 protein starts to express 4 days post infection and continues to increase to LCL level at day 28 [8,9].

## 3. EBNA2 Is the Major EBV Transcription Activator

After EBV infection, EBNALP and EBNA2 are the first two EBV genes expressed [10]. EBNALP and EBNA2 mRNAs are generated by differential splicing from the same bicistronic RNA [11,12]. EBNA2 is absolutely required for EBV to immortalize RBLs [4,13,14]. EBNA2 is the major EBV latency transcription activator [15,16] and strongly activates the EBV Cp promoter that drives the expression of all the EBNAs and the LMP1 promoter [16,17]. EBNA2 also upregulates the expression of many cell genes including MYC [18,19,20,21,22,23,24].

There are two types of EBV, type I or II. Type I EBNA2 from B95.8 strain has 487 amino acids (AAs) [25]. The EBNA2 N-terminus has two dimerization/transcription activation domains separated by a stretch of proline residues [26,27,28]. EBNA2 AAs 323–324 bind cell protein RBPJ [29,30]. The EBNA2 C-terminal region contains the major transactivation domain [31]. The RBPJ binding and the transactivation domains are required for EBV transformation [32]. Type II EBNA2 lacks the long stretch of proline repeats at the N-terminus [14]. Recombinant EBV with type II EBNA2 has lower transforming efficiency compared with virus with type I EBNA2 [14]. The reduced transformation activity is caused by S442 in the type II EBNA2 transactivation domain. Substituting the serine residue with aspartic acid that is in the type I EBV AA sequence increases the type II EBNA2 activity [33]. Increased binding to transcription repressor BS69 is also important for EBNA2 transformation activity [33,34,35]. EBNA2 is tethered to host DNA through interactions with host transcription factors (TFs). RBPJ is a major cell protein that tethers EBNA2 to DNA [29,30,36]. In T cells, RBPJ tethers Notch intracellular domain to DNA [37]. RBPJ is a transcription repressor that recruits other repressors to DNA and EBNA2 can mask the repressors to activate gene expression [38,39]. EBNA2 also increases RBPJ DNA binding [40]. EBNA2 interacts with other host TFs that can tether EBNA2 to DNA. Mutation of the SPI1 site or EBF site of the LMP1 promoter greatly reduces EBNA2 activation of the reporter [19,41]. EBNA2 binds to SPI1 and EBF and enhances EBF DNA binding [42,43]. The C-terminal transactivation domain recruits basal transcription factors including EP300/CBP, SNF/SWI complex components, TFIIH subunits, TFIIB, and TFIIE to activate transcription [44,45,46,47,48,49]. In addition, EBNA2 binds to NPM1 and PRMT5 to enhance transcription activation [50,51].

Chromatin Immune Precipitation followed by deep sequencing (ChIP-seq) identifies thousands of EBNA2 binding sites in LCLs or BL cell lines [19,52,53]. EBNA2 mostly binds to enhancers. EBNA2 binding sites overlap extensively with RBPJ binding sites. Furthermore, EBNA2 sites overlap with other B cell TFs, including EBF, ETS family TFs, RUNX3, and NF-κB family TFs in LCLs. The centers of EBNA2 binding sites have much lower H3K4me1, a histone modification marker for active promoters and enhancers, than the regions immediately adjacent to the binding site. Interestingly, these sites in RBLs are in a similar pattern, though at smaller elevation. These data suggest that EBV exploits the preexisting TF binding sites to activate host gene expression [19]. EBNA2 binds to enhancers ~556~428 kb upstream of MYC. Fluorescence in situ hybridization (FISH) and chromatin conformation capture followed by qPCR (3C-qPCR) find that these enhancers interact with MYC TSS by looping out the intervening DNA sequences between enhancers and promoters [19] (Figure 1). EBNA2 conditional inactivation greatly decreases the enhancer–promoter interaction. RNA POLII Chromatin Interaction Analysis by Paired-End Tag Sequencing (ChIA-PET) identifies all genomic interactions mediated by RNA POLII, including enhancer–enhancer, enhancer–promoter, and promoter–promoter interactions. LCL RNA POLII ChIA-PET links all the EBNA2 enhancers to their direct target genes genome-wide [54]. Most importantly, ChIA-PET links EBNA2 enhancers to MYC TSS, in agreement with data generated by capture genome-wide 3C followed by deep sequencing (C Hi-C) and circular 3C followed by deep sequencing (4C-seq) [55]. The EBNA2 enhancer interaction patterns are extremely complicated. These enhancers are found both upstream and downstream of their direct target genes. Some enhancers interact with only one target gene. Some genes interact with multiple enhancers. Some enhancers interact with multiple genes. Some enhancers skip the nearest gene to interact with genes further away. CRISPR disruption of RBPJ sites within the EBNA2 enhancer greatly reduces the mRNA levels of EBNA2-enhancer-linked genes [56].

Super-enhancers (SEs) are clusters of enhancers with extraordinary broad and high active histone marks, including H3K27ac and H3K4me1 or TF ChIP-seq peaks [57]. SEs play critical roles in development, differentiation, and oncogenesis [57]. EBNA2 SEs are enhancers with extraordinary broad and tall EBNA2 ChIP-seq peaks [54]. EBNA2 SEs control the expression of cell genes that are critically important for cell proliferation, including MYC, MAX, RUNX3, BCL2, EBF, BATF, etc. [54,58]. CRISPR deletion of MYC SE using paired sgRNAs targeting the edges of EBNA2 -525 SE greatly reduces MYC expression and cell growth in LCLs [56]. In addition to EBNA2 binding, EBNA2 SEs are also bound by many B cell TFs, including BRD4, EP300, RNA POLII, BATF, EBF, IRF4, etc. [54]. The enrichment of activated and basal TFs in SEs facilitates the phase separation that allows the formation of condensates that robustly increase the transcription activity [59,60].

## 4. EBNALP in Transcription Regulation

The AA sequence of EBNALP consists of various numbers of identical W repeats and unique Y exons in different EBV isolates. Two copies of W repeats are the minimum for EBNALP to function [61]. Four copies of W repeats are enough for EBNALP to exert optimum activity [62]. A genetic study inserting a stop codon to the end of the last W repeat indicated that EBNALP is important for EBV transformation of RBLs [63]. Mutant EBV with stop codons inserted into every W exon fails to transform naïve B cells from the cord blood [64], indicating that EBNALP is essential for EBV transformation of naïve B cells. This mutant EBV transforms memory B cells with reduced efficiency [4,64].

EBNALP increases the expression of cell cycle progression protein CCND2 [65,66]. EBNALP strongly coactivates EBNA2 transcription [62,67]. EBNALP lacks a transactivation domain in its AA sequence. It regulates transcription through removing transcription repressors such as NCOR and HADC4 from the promoters/enhancers it co-activates with EBNA2 and shuttling Sp100 between different subnuclear localizations [40,68,69]. EBNALP also greatly increases EP300 transcription activity when EP300 is tethered to DNA through the papilloma virus E2 DNA binding domain [70]. EBNALP coactivation is mostly through the EBNA2 N-terminal and C-terminal transactivation domains. EBNALP expressed in bacteria also binds to the EBNA2 transactivation domain [27]. Evolutionary conserved residues in the W repeats of EBNALP are important for coactivation [61,71]. EBNALP also binds to many cell proteins, including HA95, HAX1, HSP72, etc. to affect transcription [72,73,74].

EBNALP does not bind to DNA directly, instead it is tethered to DNA through host proteins. EBNALP ChIP-seq finds that EBNALP binds to thousands of promoter and enhancer sites [75]. Interestingly, ~33% of EBNALP binding sites are at promoters. In contrast, only ~14% of the EBNA2 binding sites are at promoters [19]. EBNALP peaks are also different from EBNA2 peaks that are very sharp and tall. EBNALP peaks tend to be much wider and lower. Integrating the ENCODE GM12878 ChIP-seq data with EBNALP data finds that 82% of the EBNALP peaks overlap with DPF2 ChIP-seq peaks. DPF2 binds to H3K14ac and H4K16ac and can bridge EBNALP to chromatin [76]. Motif analysis identifies that the motifs enrich at the EBNALP ChIP-seq peaks, including CTCF, ETS, IRF4, YY1. A total of 59% of EBNALP peaks overlaps with YY1 peaks in LCLs [75]. YY1 also binds preferentially to promoters, and it bridges promoter–enhancer interactions [77]. The high degree of overlap between EBNALP and YY1 peaks in LCLs suggests that EBNALP may play important role in mediating host genome organization (Figure 1).

## 5. EBNA3A and EBNA3C

EBNA3A, 3B, and 3C are located in tandem in the EBV genome. They all have similar gene structure with a short exon and a long exon. It is believed that they are the products of gene duplication. They share limited homology in their N-terminal AA sequences that mediate their interactions with host TF RBPJ that tethers EBNA2 to DNA [78,79,80]. Genetic studies indicate that EBNA3B is dispensable while EBNA3A is important and EBNA3C is essential for LCL growth [4,81,82,83,84]. EBNA3A AAs 170–240 and 300–523 are important for LCL growth [85]. EBNA3C AAs 50–400 and 800–900 are essential for LCL growth [86,87].

EBNA3A specifically interacts with RBPJ and prevents RBPJ binding to its cognate DNA in vitro, thus repressing EBNA2-RBPJ transcription activation [78,79,88,89]. EBNA3A-RBPJ interaction is required for EBNA3A to support LCL growth. The EBNA3A mutant, defective for RBPJ binding, fails to rescue EBNA3A inactivation in LCLs [85]. Conditional overexpression of EBNA3A induces LCL growth arrest and reduced EBNA2- RBPJ association [90]. When EBNA3A is fused to a GAL4 DNA binding domain, the fusion protein represses a reporter driven by GAL4 binding sites [91]. EBNA3A also interacts with CTBP and WDR48/USP46/USP12 deubiquitination complexes [92,93]. Transcription profiling of wild-type and EBNA3A mutant LCLs identifies many host genes repressed or activated by EBNA3A [94]. EBNA3A represses the expression of BCL2L11(BIM), CXCL9, CXCL10, MIR221/222, p16^INK4A^, p14^ARF^, CDKN2B, CDKN1A, and CDKN2C [95,96,97,98,99,100,101]. shRNA knockdown of p16^INK4A^ and p14^ARF^ allows LCLs to grow in the absence of EBNA3A [100]. EBNA3A represses BCL2L11 expression through CpG methylation and increased H3K27me3 repressive histone modification by polycomb repressive complex (PRC) proteins SUS12 and BMI1 [102,103]. EBNA3A increases H3K27me3 at the p16^INK4A^ and p14^ARF^ loci and the repression requires the interaction of EBNA3A with CTBP [104]. EBNA3A is also required for MCL-1 mitochondrial localization and activates BFL-1 transcription [105]. EBNA3A synergizes with LMP1 to induce B cell lymphoma formation in LMP1 transgenic mice and inhibits differentiation [106]. Genome-wide, EBNA3A binds to thousands of enhancer and promoter sites including MYC, CCND2, p16^INK4A^, p14^ARF^, and BCL2 [107,108,109]. Around 50% of EBNA3A sites overlap with EBNA3C sites, and 65% and 63% of EBNA3A sites overlap with BATF and RUNX3 sites. EBNA3A is tethered to DNA through BATF, shown by ChIP-re-ChIP [107].

Repression of p16^INK4A^ is the most important function of EBNA3C. shRNA knockdown of p16^INK4A^ and p14^ARF^ allows LCLs expressing conditional EBNA3C to grow in the non-permissive condition. EBNA3C inactivation does not affect the DNA methylation profiles at the loci but increases H3K27me3 and decreases H3K4me1 [84,100]. EBNA3C null virus can transform B cells from primary B cells from an individual with homozygous deletion of p16^INK4a^ [110]. CRISPR knockout of p16^INK4A^ also allows LCLs to grow in the absence of EBNA3C [111]. EBNA3C binds to p14^ARF^ promoter and recruits Sin3A repressor to the site. The Sin3A recruitment is dependent on EBNA3C expression [112]. EBNA3C interactions with RBPJ, CtBP, WDR48, and USP46 are also important for EBNA3C-mediated p16^INK4A^ repression [92,93,111] (Figure 1).

EBNA3C maintains the LMP1 expression level in growth-arrested Raji cells [113]. EBNA3C co-activates the LMP1 promoter together with EBNA2 [114,115]. EBNA3C upregulates the expression of AID that is important for somatic hypermutation and class switch [116,117]. EBNA3C upregulates CXCL12 and CXCR4 expression in an RBPJ-dependent way [117]. EBNA3C represses the expression of BCL2L11(BIM) by recruiting PRC components SUZ12 and EZH2 to the regulatory elements and enhancer reorganization [55,118]. EBNA3C recruits RBPJ, SUZ12, and BMI1 to repress ADAMDEC1 and ADAM28 [119]. EBNA3C also induces the expression of miRNAs miR-221/222 [99].

Genome wide, EBNA3C binds to thousands of enhancers and promoters in LCLs [108,112]. EBNA3C binding sites coincide with RUNX3, BATF, ATF2, and IRF4. EBNA3C signals are strongest at BATF/IRF4 and SPI1/IRF4 composite sites. EBNA3C binds strongly to the p14^ARF^ promoter through SPI1/IRF4/BATF/RUNX3 host TFs and recruits transcription repressors Sin3A and REST family members in an EBNA3C-dependent manner [112].

EBNA3C specifically interacts with RBPJ and the interaction is required for continuous LCL growth [79,80,86,87,120]. The EBNA3C N-terminal homology domain binds to RBPJ. EBNA3C prevents RBPJ from binding to its cognate sites in vitro, but it can also recruit RBPJ to DNA at some sites in vivo. However, the genome-wide effect of EBNA3C on RBPJ DNA binding is still not known [79,88,119]. EBNA3C interacts with IRF4 and IRF8 through its N-terminus and stabilizes IRF4 [121]. IRF4 is essential for LCL growth [122] and may tether EBNA3C to DNA [112]. EBNA3C promotes proteasome-mediated degradation of IRF8 [121]. EBNA3C specifically modulates cell-cycle-regulating proteins. EBNA3C amino acids 130 to 159 bind CCNA and enhance CCNA-dependent kinase activity [123,124]. These amino acids are required for LCL growth [86,87,125]. EBNA3C stabilizes CCND2 to regulate cell cycle progression [126]. EBNA3C also stabilizes CCND1 through inhibition of its polyubiquitination and EBNA3C enhances the kinase activity of Cyclin D1/CDK6 [127] (Figure 1). EBNA3C amino acids 130–160 bind to two different sites on CCND1. EBNA3C associates with the CCND1/cdk2 complex and enhances the kinase activity. EBNA3C recruits SCF^Skp2^ activity to CCNA complexes [128]. Through Skp2, EBNA3C also degrades tumor suppressor RB [129]. EBNA3C amino acids 140–149 are important for both the binding and regulation of RB [129]. EBNA3C inhibits the function of p53 through interaction with ING4/5 [130]. EBNA3C can modulate the PIM1 kinase activity [131]. EBNA3C amino acids 365 to 545 are important for EBNA2 coactivation of the LMP1 promoter and interact with SUMO-1 and SUMO-3 [132]. EBNA3C recruitment of histone deacetylase activity and association with the corepressors Sin3A and NCoR are likely to be important for EBNA3C-mediated transcription repression [133].

## 6. EBNA1

EBNA1 is expressed in all EBV-infected cells. EBNA1 is essential for EBV genome replication, persistence, and transcription [134,135,136,137,138,139]. EBNA1 tethers EBV episomes to the mitotic chromosomes during mitosis to ensure even distribution of episomes into daughter cells [136]. ChIP and microarray identified EBNA1 binding sequences in vitro [140]. ChIP-seq identified EBNA1 binding sites in vivo [141,142]. EBNA1 binding to MEF2B, IL6R, and EBF1 is critical for B cell survival [142]. During EBV transformation of RBLs to LCLs, EBNA1 binds to enhancers linked to genes involved in nucleotide metabolism [143]. EBNA1 forms a DNA protein crosslink with the EBV origin of replication oriP [144]. The protein–DNA crosslinking enables OriP replication termination to promote episome maintenance [144]. A small molecule inhibitor, that selectively inhibits EBNA1 DNA binding, prevents EBV infection and tumor growth [145]. This EBNA1 inhibitor also inhibits gastric cancer growth [146]. Another EBNA1 small molecule has also been reported to perturb episome replication and persistence [147].

## 7. LMP1

LMP1 activates canonical and non-canonical NF-κB pathways [148]. NF-κB family TFs have five subunits that can form a heterodimer or homodimers and bind to DNA [149]. These proteins are each critical for B cell development and function. NF-κB subunit ChIP-seq identifies a complex NF-κB-binding landscape in LCLs. Nearly one-third of NF-κB-binding sites lack κB motifs and are instead enriched for alternative motifs. The oncogenic forkhead box protein FOXM1 co-occupies nearly half of NF-κB-binding sites and forms protein complexes with NF-κB on DNA. FOXM1 knockdown decreases NF-κB target gene expression and ultimately induces apoptosis, highlighting FOXM1 as a synthetic lethal target in B cell malignancy [149].

## 8. Future Directions

The expression of viral transcription factors in EBV cancer provides a unique opportunity for targeted therapies. Traditionally, TFs are not ideal drug targets. However, the development of proteolysis targeting chimeras (PROTACs) now allows specific degradation of TFs [150]. A small molecule that binds to TFs is linked to E3 ligase ligand. The molecule then recruits E3 ligase to the TF and degrades the TF. PROTACs have been developed to target many TFs and the degradation of TFs important for cancer cells can inhibit cell growth [151,152,153]. It is therefore now possible to develop EBV-specific PROTACs to degrade EBV TFs to treat EBV-associated cancers.

## Figures and Tables

**Figure 1 viruses-15-00832-f001:**
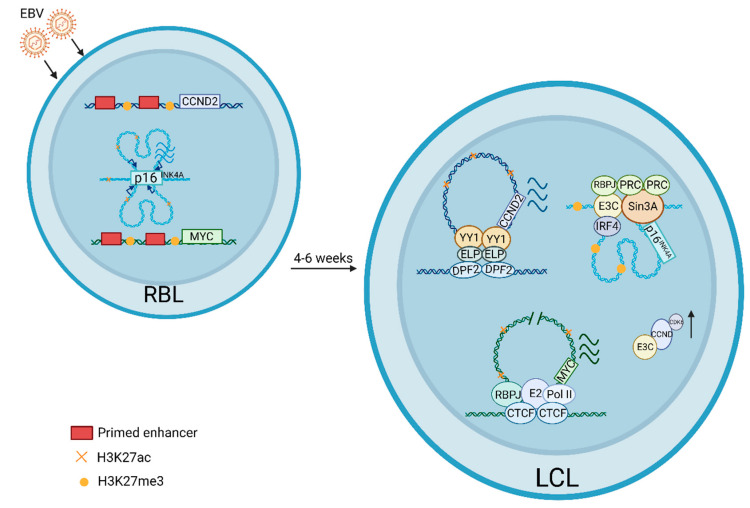
Host enhancer/silencer rewiring following EBV transformation of RBLs into LCLs. In RBLs, MYC and CCND2 are expressed at basal level with few enhancer promoter interactions. p16INK4A is expressed at high level with extensive enhancer–promoter interactions and active enhancer mark H3K27ac. Following EBV transformation, EBNA2 induces enhancers 400–500 kb upstream of MYC to loop to MYC TSS to activate MYC expression. EBNALP recruits YY1 to CCND2 locus and promotes enhancer–promoter interaction to activate CCND2 expression. These loci are now marked by H3K27ac. At the p16INK4A locus, much less enhancer–promoter interactions is seen and EBNA3C recruits transcription repressors Sin3A and polycomb repressive complexes to the locus with increased H3K27me3. EBNA3C stabilizes CCND and increases CDK6 kinase activity.

## Data Availability

Not applicable.

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
