# Peer review of "Epstein–Barr Virus B Cell Growth Transformation: The Nuclear Events"

_viruses, 2023, doi:10.3390/v15040832_

Round 1

Reviewer 1 Report

This exhaustive and well-written review provides a complete view of the significant transcriptional events that occur after infection of B cells by the Epstein-Barr Virus. The author describes the transcriptional functions of relevant viral proteins in detail, providing a state-of-the-art view of how EBV hijacks host transcription. This review will be of high interest to the virology field. 

Author Response

Thank you for the positive comments. Spelling has been checked.

Reviewer 2 Report

Epstein-Barr virus (EBV) is associated with many cancers worldwide. In vitro, it immortalizes human resting-B lymphocytes to Lymphoblastoid cell lines (LCLs). These LCLs constitute a pertinent model to study the mechanisms by which EBV infection leads to cell growth transformation.

In this review, the author focus on the main nuclear steps of this process. Five nuclear viral proteins are required to achieve EBV-mediated cell immortalization. The author give a very extensive overview of the various mechanisms by which these proteins, mostly transcriptional factors, participate in the immortalisation process. The review, which gathers a huge amount of information in only a few pages, is overall well written but very dense. It would benefit from the addition of one or two Figures that would summarize the main events described in the review.

Specific comments:

- There is a few typo mistakes to be corrected:

Line 12: activated instead of activates

Line 31: one o is missing in lymphoproliferative

Line 69:  Masks instead of mask

Line 76: enhancer instead of enhance

Line 116: of missing after consists

Line 129: EBANLP instead of EBNALP

Line 130-131: a verb is missing in the sentence

Line 196 : EBAN3C instead of EBNA3C

Line 200: a space is missing between the and p14

Line 208: To is missing before DNA

Line 219: kniase instead of kinase

- Line 158: The authors should also cite the reference

Waltzer L, Perricaudet M, Sergeant A, Manet E. J Virol. 1996 Sep;70(9):5909-15. Epstein-Barr virus EBNA3A and EBNA3C proteins both repress RBP-J kappa-EBNA2-activated transcription by inhibiting the binding of RBP-J kappa to DNA.

Author Response

Thank you for the comments. All comments are now addressed and spelling checked.

Reviewer 3 Report

The manuscript entitled “Epstein-Barr virus B cell growth transformation: the nuclear 2 events” reviews the impact of key viral proteins on infected B lymphocytes. The topic is important although the author has not clarified what is aim of this manuscript. Along this line, as there are several review papers on EBV infection, what is the novelty of this manuscript? Additionally, a Figure (s)  complementing the main text focusing on the function of these key proteins is missing.

Author Response

Thank you for the comments. This review includes some unpublished data from my lab. A figure is now included. The spelling has been checked.